# “Healthcare Kamikazes” during the COVID-19 Pandemic: Purpose in Life and Moral Courage as Mediators of Psychopathology

**DOI:** 10.3390/ijerph18147235

**Published:** 2021-07-06

**Authors:** Iván Echeverria, Marc Peraire, Gonzalo Haro, Rafael Mora, Isabel Camacho, Isabel Almodóvar, Vicente Mañes, Ignacio Zaera, Ana Benito

**Affiliations:** 1TXP Research Group, Universidad Cardenal Herrera-CEU, CEU Universities, 12006 Castelló de la Plana, Spain; gomechiva@alumnos.uchceu.es (I.E.); perairemiralles@gmail.com (M.P.); gonzalo.haro@uchceu.es (G.H.); isabel.almodovar@uchceu.es (I.A.); 2Department of Mental Health, Consorci Hospitalari Provincial de Castelló, 12002 Castelló de la Plana, Spain; rafael.mora@hospitalprovincial.es (R.M.); vijomavi@hotmail.com (V.M.); izcizc@hotmail.com (I.Z.); 3Department of Mental Health, Hospital Psiquiátrico de Campeche, Campeche 24560, Mexico; coquet_@hotmail.com; 4Torrent Mental Health Center, Hospital General de Valencia, 46900 Torrent, Spain

**Keywords:** acute stress, anxiety, COVID-19, coronavirus, depression, psychopathology, purpose in life, moral courage, healthcare workers, healthcare students

## Abstract

Although the required personal protective equipment was not available during the first wave of the COVID-19 pandemic, Spanish healthcare workers continued to work, being dubbed as ‘healthcare kamikazes’. Two possible reasons are moral courage and purpose in life that, in turn, would modulate the appearance of psychopathology. Cross-sectional study was carried out in 90 Spanish and 59 Mexican healthcare professionals, and 56 medical and nursing students. Spanish professionals had suffered more work and overall exposure (M = 8.30; SD = 2.57 and M = 9.03; SD = 2.66) than Mexican (M = 5.10; SD = 1.87 and M = 5.55; SD = 2.35). Mexican professionals had fewer anxiety disorders (30.5%; *n* = 18) and a lower depression score (M = 4.45; SD = 5.63) than the Spanish (43.7%; *n* = 38; and M = 8.69; SD = 8.07). Spanish professionals more often experienced acute stress disorder (32.6%; *n* = 29). Purpose in life, in addition to having a direct protective effect on psychopathology, also modulated the relationship between personal and family exposure and psychopathology. In conclusion, purpose in life protects against the appearance of psychopathology in healthcare workers with personal and family exposure to SARS-CoV-2.

## 1. Introduction

On 31 December 2019, an outbreak of 27 cases of pneumonia of unknown etiology was reported in Wuhan, Hubei province, China. The causative agent of this pneumonia turned out to be a new coronavirus, SARS-CoV-2, with the clinical picture caused by this virus being denominated COVID-19. Since the WHO declared a global pandemic on March 11, 2020, there have been 107 million confirmed cases worldwide, with more than three million cases in Spain. Because of the unexpected nature of the COVID-19 pandemic, the Spanish health system was overwhelmed, leading to moments when there were shortages of personal protective equipment (PPE) such as masks, disposable gowns, protective screens, etc. Consequently, up to 26% of all the people infected by coronavirus in Spain during the first wave were healthcare professionals compared to around 9% in other countries such as Italy [1]. Despite the inherent risk, healthcare workers did not stop providing care, which is why the New York Times dubbed them ‘healthcare kamikazes’ [2].

One of the reasons for this phenomenon could be ‘moral courage’, a term that refers to the ability to face danger or social disapproval when performing what one believes to be their duty [3]. Along with this, purpose in life (PIL), that is, an individual’s perception of the objective and value of their life, is another aspect that can motivate professionals to decide to continue working, despite the risk to their lives. Paradoxically, the same moral courage that makes these professionals determined to go to work for their patients, is a factor that can put their mental health at risk because anxiety or depression may be experienced when it is impossible to act according to these moral values (i.e., moral distress) [4]. In turn, various studies have shown the relationship between decreased PIL and the appearance of anxiety, depression, and substance abuse, as well as the protective role of high levels of PIL in the appearance of these symptoms [5,6].

In addition to the risk of contagion, there are also precedents for the impact epidemics or pandemics can have on the mental health of healthcare workers, even one year after their resolution. For example, high levels of stress, depression, anxiety, and post-traumatic stress symptoms were registered during the SARS epidemic in 2003 [7,8,9]. Moreover, various studies have already evaluated the impact COVID-19 has had on healthcare workers. A Chinese study [10], showed that up to 50.4% of the sample presented symptoms of depression, 44.6% showed signs of anxiety, and 34% had insomnia, with the risk of suffering these symptoms being higher among healthcare workers who worked in direct contact with COVID-19 patients. Another study [11] assessed the severity of these aforementioned symptoms, noting that 34.4% of the sample had mild symptoms, 22.4% presented moderate symptoms, and 6.2% showed severe symptoms. A study carried out in Spain indicated that 22.5% of healthcare workers met the criteria for generalized anxiety disorder, while 28.1% had major depressive disorder, 22.2% presented post-traumatic stress disorder, and 6.2% had a substance use disorder; thus, a total of 45.7% of the sample displayed some ongoing mental disorder [12].

Although some studies have registered the presence of mental disorders such as anxiety, depression and post-traumatic stress, during some epidemiological emergencies, none have related the prevalence of the appearance of psychopathology or the type of psychopathology to motivational factors such as PIL or moral courage in this context. Therefore, this current work aimed to determine the prevalence of psychopathology derived from exposure to the COVID-19 pandemic in healthcare workers and to assess whether moral courage or PIL behave as protective or predisposing factors for any of these psychopathological complications.

## 2. Materials and Methods

This was an observational, cross-sectional study of cases and controls. A group of 90 Spanish healthcare workers (doctors, nurses, nursing assistants, administrative staff, security members, cleaning personnel, psychologists, social workers, and orderlies) and another group of 59 Mexican healthcare workers were obtained through convenience sampling. Individuals in the Spanish sample had worked at the Consorci Hospitalari Provincial de Castelló which is the second biggest hospital in the city, responsible, among others, for the mental health, oncology, and ophthalmology departments in the province of Castelló, or in other centers dependent on the Castelló Health Department during the COVID-19 pandemic, while the Mexican sample had worked at the Hospital Psiquiátrico de Campeche, the main mental health institution in the province of Campeche, and other health centers in the same city. A sample of 56 medical and nursing students who were in their final year at university, also obtained by convenience sampling, were used as a comparison group.

We used G*Power software (v3.1.9.4) [13] to calculate that a sample size of 198 would be required when considering an expected effect size f 2 (V) of 0.0625, an alpha of 5%, and beta of 20% for the 3 groups, with 13 response variables and when performing MANOVA global effects analysis. The data on Spanish workers and students were obtained between 20 April and 27 May 2020, when Spain was at the peak of its first wave of COVID-19, while the information on Mexican workers was collected between 27 May and 13 June of the same year while Mexico was at the beginning of the pandemic.

After signing their informed consent to participation, the study participants performed a self-evaluation using a series of instruments. First, they completed a questionnaire on sociodemographic variables. The different types of exposure to COVID-19 were considered as independent variables. To assess personal and family exposure, we employed an ad hoc questionnaire, and for occupational exposure, an ad hoc scale based on the classifications of the Spanish Ministry of Universal Health Service and Public Health was used. The different job types of the individuals in the cohort were also classified into areas according to the impact the infection had had in each one. The total exposure was calculated by summing personal, family and occupational exposure. Variables that evaluated psychopathology were considered dependent variables.

To assess anxiety, depression, and acute stress we used the Beck Anxiety Inventory (BAI; cut-off point [CP] = 8) [14], Beck Depression Inventory (BDI-II; CP = 14) [15], and an ad hoc questionnaire based on the DMS-5 criteria for assessing acute stress, respectively. Drug abuse was assessed using the Drug Abuse Screening Test (DAST-10; CP = 1) [16] and alcohol abuse was tested using the Alcohol Use Disorders Identification Test (AUDIT; CP for women = 6, CP for men = 8) [17]. Total scores and dichotomous variables were calculated to divide the participants into individuals whose score exceeded the scale CPs and those where it did not.

The purpose in life, moral courage, and strength of character were considered as modulating variables. The purpose in life was analyzed using the Purpose in Life (PIL) scale (CP = 113) [18], calculating a dichotomous variable to differentiate between individuals who had a sense of PIL and those who did not. Moral courage was assessed with the Moral Courage Scale for Physicians (MCSP) [19] and the Professional Moral Courage Scale (PMCS) [20]. Finally, the Global Assessment of Character Strengths-24 (GACS-24) [21] was used to assess the strength of character of the participants.

SPSS software (version 23) for Microsoft (IBM Corp., Armonk, NY, USA) was used for all the statistical analyses. After the exploratory and descriptive study, the quantitative variables were compared using MANOVA given that several of them correlated with each other. The variables were subsequently compared using ANOVAs for quantitative variables and Pearson chi-squared tests for categorical variables. Linear and logistic regression models were created for the psychopathological variables, introducing exposure to SARS-CoV-2 and modulating variables as independent variables. Finally, the data were modeled using PROCESS v3.4 for SPSS [22] to test two hypotheses: (1) personal and family exposure increases anxiety, depression, acute stress, and total psychopathology; and (2) purpose in life has a buffering effect, modulating the relationship between personal and family exposure and psychopathology.

The ethical principles set out in the Declaration of Helsinki and the Council of Europe Convention were followed and the informed consent of all participants was obtained. Moreover, data confidentiality was guaranteed according to the General Data Protection Regulation (GDPR; 2018). This study was authorized by the Investigation Commission at the Provincial Hospital Consortium in Castellon (ref. A-15/04/20) and the Clinical Research Ethics Committee of the Cardenal Herrera-CEU University (ref. CEI20/068).

## 3. Results

### 3.1. Sociodemographic Characteristics of the Sample

Table 1 shows the sociodemographic characteristics of the sample and the comparisons between the groups. Both the mean student age (M = 24.7; SD = 3.8) and mean Mexican professional age (M = 40.7; SD = 8.1) was lower than that of the Spanish professionals (M = 44.8; SD = 10.7). The Spanish professionals included more women (73.3%; *n* = 66) than men (26.7%; *n* = 24). Mexican professionals presented a higher percentage of religiosity than Spanish professionals (83.1%; *n* = 49). Regarding marital status, students tended to be single (94.6%; *n* = 53), a higher percentage of Mexicans were divorced (16.9%; *n* = 11) and a lower percentage were single (27.1%; *n* = 16); more Spanish professionals were married (61.1%; *n* = 55) rather than single (25.6%; *n* = 23).

Regarding educational level, logically, 100% of the students were pursuing a university degree, while most professionals had a university degree (77.6% for Mexican professionals and 75.6% for Spanish professionals). More Mexican professionals had not finished secondary education or an equivalent level of education (12.1%; *n* = 7) while more Spanish professionals had finished (14.4%; *n* = 13) and had not completed a university degree (7.8%; *n* = 7). Most students had studied in a private center (96.4%; *n* = 54), while most of the professionals had studied in a public center (78.0% Mexican versus 81.2% Spanish). Of the three groups, students presented the lowest percentage of physical illness (14.3%, *n* = 8). Finally, Mexican professionals (M = 0.2; SD = 0.7) smoked more cigarettes than Spanish professionals (M = 2.7; SD = 5.9). MANOVA analyses indicated that there were differences between the groups in terms of the study variables (F = 12.429; *p* < 0.001; ES = 0.487; [1−β] = 1).

### 3.2. Exposure to SARS-CoV-2, Modulating Variables and Psychopathology

Table 2 shows the scores of the participants for the independent, modulating, and dependent variables. There were no differences in personal and family exposure to SARS-CoV-2; students had lower total and occupational exposure (M = 0; SD = 0 and M = 0.62; SD = 0.90) than Spanish professionals (M = 8.30; SD = 2.57 and M = 9.03; SD = 2.66), who in turn had suffered more work and overall exposure than Mexican professionals (M = 5.10; SD = 1.87 and M = 5.55; SD = 2.35).

Mexican professionals had greater moral courage (M = 11.1; SD = 0.93) than Spanish professionals (M = 10.6; SD = 1.36) and students (M = 10.48; SD = 1.53) on the PMCS scale. They also had higher PIL scores (M = 123.3; SD = 12.6) than Spanish professionals (M = 109.0; SD = 14.9) and students (M = 109.30; SD = 14.1). When the CP was used, Mexican professionals presented higher percentage of PIL (79.7%; *n* = 47) than Spanish professionals (43.8%; *n* = 39). Mexican professionals also presented greater GACS-24 (M = 145.7; SD = 16.9) than the Spanish professionals (M = 132.9; SD = 16.7) and students (M = 133.2; SD = 15.6).

Regarding psychopathology, there were no differences in the mean BAI score, but when the CP was used, students presented higher percentage of anxiety disorder (53.6%; *n* = 30) than Mexican professionals (30.5%; *n* = 18). Mexican professionals had lower BDI-II scores (M = 4.45; SD = 5.63) than Spanish professionals (M = 8.69; SD = 8.07) and students (M = 8.80; SD = 5.82). However, when the CP was used, there was no difference in the proportion of participants with depressive disorder. Spanish professionals presented more acute stress (M = 8.59; SD = 7.70) than Mexican professionals (M = 5.71; SD = 5.49) and students (M = 6.07; SD = 3.61), and when the CP was used, Spanish professionals showed a higher proportion of acute stress disorder (32.6%; *n* = 29).

There were no differences between the groups for the DAST-10 score for the presence of drug use disorder. However, there were differences in the AUDIT score with a trend towards higher scores among students than in professionals (M = 3.51; SD = 3.28), although this difference did not reach significance in post-hoc tests. There were no significant differences in the presence of alcohol use disorder. The students had more psychopathological symptoms and more mental disorders (M = 28.42; SD = 14.65 and 69.6%, respectively; *n* = 39) than the Mexican professionals (M = 20.30; SD = 17.89 and 39%, respectively; *n* = 23).

Table 3 shows the results of the linear regressions that allowed us to predict the presentation of psychopathology based on the SARS-CoV-2 exposure variables and modulating variables. The BAI score could be predicted by personal and family exposure (odds ratio [OR] = 1.959; 95% CI [0.719, 3.200]); *p* = 0.002) and PIL (OR = −0.166; 95% CI [−0.274, −0.057]; *p* = 0.003). The BDI-II was predicted by personal and family exposure (OR = 1.308; 95% CI [0.351, 2.265]; *p* = 0.007), PIL (OR = −0.206; 95% CI [−0.289, −0.123]; *p* < 0.001), and GACS-24 (OR = −0.084; 95% CI [−0.156, −0.012]; *p* = 0.002). The acute stress score was predicted by personal and family exposure (OR = 1.181; 95% CI [.247, 2.116]; *p* = 0.013), occupational exposure (OR = 0.320; 95% CI [0.115, 0.525]; *p* =0.002), and PIL (OR = −0.147; 95% CI [−0.228, −0.066]; *p* < 0.001). DAST-10 could be predicted by PIL (OR = −0.009; 95% CI [−0.016, −0.002]; *p* = 0.013). AUDIT was predicted by occupational exposure (OR = −0.161; 95% CI [−.257, −0.065]; *p* = 0.001) and GACS-24 (OR = −0.039; 95% CI [−0.071, −0.006]; *p* = 0.021). Finally, overall psychopathology was predicted by personal exposure (OR = 4.995; 95% CI [2.219, 7.771]; *p* < 0.001) and PIL (OR = −0.513; 95% CI [−0.756, −0.271]; *p* < 0.001).

Table 4 shows the variables that could predict the presence of mental disorders. Of note, personal and family exposure allowed the prediction of anxiety disorder (B = 1.662; 95% CI [1.090, 2.533]; *p* = 0.018), depressive disorder (B = 1.968; 95% CI [1.235, 3.136]; *p* = 0.004), acute stress disorder (B = 1.911; 95% CI [1.227, 2.977]; *p* = 0.004), and the presence of a mental disorder (B = 1.858; 95% CI [1.203, 2.870]; *p* = 0.005). Similarly, moral courage could predict anxiety disorder (B = 1.650; 95% CI [1.102, 2.472]; *p* = 0.015), acute stress disorder (B = 1.753; 95% CI [1.097, 2.801]; *p* = 0.019), and the presence of a mental disorder (B = 1.682; 95% CI [1.151, 2.458]; *p* = 0.007). Finally, PIL allowed the prediction of depressive disorder (B = −0.931; [−0.900, −0.962]; *p* < 0.001) and acute stress disorder (B = −0.949; [−0.919, −0.980]; *p* = 0.001).

Lastly, Figure 1 shows the models that described the modulating effect PIL had on personal and family exposure to SARS-CoV-2 and psychopathology. Model 1 shows that, in addition to directly affecting anxiety, PIL also modulated the relationship between personal and family exposure and anxiety, especially at low (B = 3.78; 95% CI [2.49, 5.07]; *p* < 0.001) and moderate (B = 2.22; 95% CI [1.16, 3.28]; *p* < 0.001) values. As shown in model 2, the PIL did not modulate the relationship between personal and family exposure and depression, although PIL did directly affect depression.

Model 3 showed that personal and family exposure influenced acute stress and its effects were modulated by PIL. This modulation occurred at low (B = 3.11; 95% CI [2.11, 4.11]; *p* < 0.001) and moderate (B = 1.50; [0.676, 2.33]; *p* < 0.001) values. Finally, in model 4 we observed that, in addition to directly affecting psychopathology, PIL modulated the relationship between personal and family exposure and psychopathology, especially at low (B = 9.85; 95% CI [6.94, 12.77]; *p* < 0.001) and moderate (B = 5.66; 95% CI [3.26, 8.07]; *p* < 0.001) values.

## 4. Discussion

Multiple studies evaluating the impact of the COVID-19 pandemic on the mental health of healthcare workers have been published in recent months with findings of high levels of anxiety, depression, or acute stress [10,23,24]. However, this current study is the first work to determine the role of PIL as a modulating variable between exposure to SARS-CoV-2 and psychopathology. In agreement with other studies [6,25] showing that high PIL scores were related with the appearance of lower levels of anxiety, the results of our work showed that when facing personal and family exposure to SARS-CoV-2, high PIL reduced the appearance of anxiety. A high level of PIL reduced the emergence of depression, as also demonstrated elsewhere [25,26], but the PIL did not modulate the relationship between exposure and psychopathology. This could be because, the relationship between PIL and depression could be bidirectional, with depression leading to low levels of PIL and vice versa [27,28]. Finally, we observed that a high PIL score reduced acute stress, when confronting personal and family exposure to SARS-CoV-2, as it has been previously observed in other disaster situations [29].

These results may be explained by the fact that PIL is included in the logotherapy which holds that life can have purpose and sense even in the most impoverished circumstances, making people more resilient in terms of surviving harsh conditions. Both terms are related to the positive psychology and the salutogenesis framework, which states that people who view their life as having positive influence on their health, may stay well and even be able to improve adaptive coping in stressful situations. This framework assumes that people have resources available (biological, material, and psychosocial) that enable them to construct coherent life experiences. Failure to do so could explain their psychopathology [30]. Considering all the above, there is evidence that high levels of PIL played a protective role (the buffering model described by Cohen and Wills) [31] in the effect that personal and family exposure to the virus had on overall pathology, anxiety and acute stress.

Regarding moral courage, high scores increased the probability of presenting anxiety disorders and acute stress. Although no work has yet explored this relationship, this effect could perhaps be explained through the concept of moral distress [4]: by increasing the difference between an individual’s moral expectations and the behavior they are able to implement, a dissonance occurs that can lead to acute anxiety and stress. This difference would be greater among professionals with high moral courage (e.g., especially determined workers who are highly invested in their patients), who see that they have insufficient means (e.g., personal protective equipment, staff, etc.) to carry out their work to the level of excellence they seek, causing the professional to be prevented from acting in the way he or she considers correct and increasing their psychopathology [32].

Despite their similarities, intrinsic factors that determine PIL and moral courage are not the same: while PIL depends on life perception and goals, moral courage is influenced by ethical principles and moral obligation, which would explain the differences between both and the derived psychopathology.

Our study also revealed the psychopathological differences between Spanish and Mexican professionals, with the latter presenting better mental health and more positive character aspects. This may be because the Spanish cohort was more heterogeneous in terms of the types of jobs it included, while practically all the individuals evaluated in the Mexico group worked in mental health. Bearing this in mind, two possibilities should be considered: (1) that, mental health workers may pay greater attention and take greater care of their personal mental health because of their knowledge and skills in this field; and (2) they provided more socially desired answers because they may have known which variables were positive or negative on the survey scales we used. To date, no other studies have assessed whether mental health workers present differential characteristics in terms of mental self-care compared to other healthcare workers. However, there is research that supports the hypothesis that individuals that present greater social desirability bias tend to minimize psychopathologies [33].

In contrast, Spanish healthcare workers presented significantly higher levels of acute stress than students and Mexican professionals. This result could be explained by the fact that the Spanish sample was evaluated in the middle of the first peak in the COVID-19 pandemic, while the Mexican sample was evaluated when the pandemic was just starting to affect the country, meaning the latter group were likely to present less psychopathology related to the stressful events occurring at the time. In addition, the high religiosity of Mexican workers compared to the Spanish professionals and students might imply higher PIL scores and therefore, a lower risk of the appearance of psychopathology.

The high prevalence of mental disorders in students, which was similar to that of Spanish professionals, may have been caused by the psychological and emotional impact the pandemic had had on them, which would have been aggravated by anxiogenic factors such as news anticipating their incorporation as reinforcement personnel during the pandemic [34]. In addition, multiple studies have reflected the high baseline rate of mental disorders present in medical students [35,36]. This prevalence could be related to the high average scores in neuroticism and perfectionism usually obtained by medical students, in addition to their academic load, sleep deprivation, or dissatisfaction with their results, all of which can lead to poorer mental health. Along with this, the international lag in the phases of the pandemic could also perhaps explain why the students in our study had significantly higher levels of mental disorders compared to the Mexican professionals.

The main limitation of this work was its cross-sectional nature, meaning that we were unable to infer causality. Thus, we cannot state with certainty that the psychopathological results we found exclusively corresponded to the COVID-19 pandemic and the influence of modulating variables. Another possible limitation was the use of a group of students as a control group because of their sociodemographic differences (e.g., age, parenthood, etc.), which may have influenced the presence of psychopathological differences. Nonetheless, these seemed to be related to different levels of exposure and to the participant’s perceived PIL. When interpreting the results, the time lag between countries must also be considered because the survey was carried out in the midst of the first wave of the pandemic in Spain, while in Mexico it was just starting to grow exponentially.

## 5. Conclusions

In conclusion, this is the first study published to date to show the protective role of purpose in life in the appearance of psychopathology in healthcare workers, in relation to personal and family exposure to SARS-CoV-2. In this sense, it would be advisable for healthcare teams to spend part of their time (and health students to spend part of their training) strengthening their purpose in life through exercises of individual introspection and reflection upon personal values, goals, and passions [30]. However, the fact that high moral courage increases the risk of anxiety and acute stress, means that health systems should be designed and equipped with the resources required to allow professionals to properly complete their work. The results in this cross-cultural sample suggest that, although the high moral courage typical of the ‘healthcare kamikazes’ could increase the risk of anxiety and acute stress, the quality that most strongly protected healthcare workers was a strong purpose in life.

## Figures and Tables

**Figure 1 ijerph-18-07235-f001:**
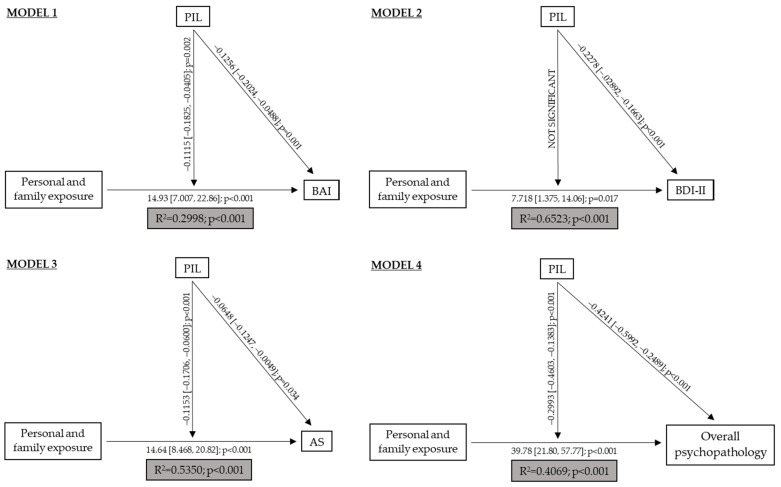
Explanatory models of psychopathology in healthcare workers. PIL: Purpose in Life; BAI: Beck’s Anxiety Inventory; BDI-II: Beck’s Depression Inventory; AS: Acute Stress.

**Table 1 ijerph-18-07235-t001:** Sociodemographic characteristics and differences between the study groups.

	Total *n* = 205	Students *n* = 56	Mexican Professionals *n* = 59	Spanish Professionals *n* = 90	
	*n* (%)/ M (SD)	*n* (%)/ M (SD)	*n* (%)/ M (SD)	*n* (%)/ M (SD)	$F/\chi^{2}$ (*p*) Post-Hoc/CTR
**AGE**	37.4 (11.9)	24.7 (3.8)	40.7 (8.1)	44.8 (10.7)	97.807 (<0.001) 0.037 (Mx < Sp) <0.001 (St < Mx) <0.001 (St < Sp)
**SEX**					6.762 (0.034)
**Female**	144 (70.2)	44 (78.6)	34 (57.6)	**66 (73.3)**	1.6/0.9/2.5
**Male**	61 (29.8)	12 (21.4)	25 (42.4)	**24 (26.7)**	−1.6/−0.9/-2.5
**RELIGIOSITY Yes**	138 (67.3)	33 (58.9)	**49 (83.1)**	56 (62.2)	9.791 (0.009) −1.6/3.1/−1.4
**MARITAL STATUS**					80.230 (<0.001)
**Single**	92 (44.9)	**53 (94.6)**	**16 (27.1)**	**23 (25.6)**	8.8/−3.2/−4.9
**Married**	87 (42.4)	**2 (3.6)**	30 (50.8)	**55 (61.1)**	−6.9/1.5/4.8
**Divorced**	22 (10.7)	**1 (1.8)**	**11 (16.9)**	10 (11.1)	−2.5/2.3/0.2
**Widowed**	4 (2.0)	0	2 (3.4)	2 (2.2)	−1.2/0.9/0.2
**EDUCATION**					211.600 (<0.001)
**University education**	113 (55.4)	**0**	**45 (77.6)**	**68 (75.6)**	−9.8/4.0/5.1
**Pursuing university degree**	56 (27.5)	**56 (100)**	**0**	**0**	14.3/−5.5/−7.8
**Did not finish university education**	8 (3.9)	0	1 (1.7)	**7 (7.8)**	−1.8/−1.0/2.5
**Secondary education/vocational training/preparatory courses**	18 (8.8)	0	5 (8.6)	**13 (14.4)**	−2.7/−0.1/2.5
**Did not finish secondary education/vocational training/preparatory courses**	9 (4.4)	0	**7 (12.1)**	2 (2.2)	−1.9/3.4/−1.4
**EDUCATION CENTER**					91.975 (<0.001)
**Private**	80 (43.5)	**54 (96.4)**	**13 (22)**	**13 (18.8)**	9.6/−4.0/−5.2
**Public**	104 (56.5)	**2 (3.6)**	**46 (78)**	**56 (81.2)**	−9.6/4.0/5.2
**PHYSICAL ILLNESS Yes**	55 (26.8)	**8 (14.3)**	20 (33.9)	27 (30.3)	6.526 (0.038) (−2.5/1.4/1.0)
**SMOKER Yes**	39 (19.1)	10 (17.9)	7 (11.9)	22 (24.7)	3.871 (0.144)
**CIGARETTES/DAY**	1.6 (4.5)	1.4 (3.8)	0.2 (.7)	2.7 (5.9)	5.591 (0.004) 0.001 (Mx < Sp)
**PSYCHIATRIC HISTORY Yes**	43 (21.0)	8 (14.3)	15 (25.4)	20 (22.2)	2.301 (0.317)
**PSYCHOLOGICAL OR PSYCHOPHARMACOLOGICAL TREATMENT DURING QUARANTINE Yes**	15 (7.4)	2 (3.6)	8 (13.6)	5 (5.6)	4.905 (0.086)

CTR: corrected typified residuals; those less than −1.96 or greater than 1.96 were considered significant. The groups from among the categorical variables in which the CTRs were significant are shown in bold. When the post-hoc tests in the quantitative variables were significant we have indicated between which groups the differences occurred. St: Students; Mx: Mexican professionals; Sp: Spanish professionals.

**Table 2 ijerph-18-07235-t002:** Scores in the independent, modulating, and dependent variables and differences between the groups.

	Total *n* = 205	Students *n* = 56	Professionals Mexican *n* = 59	Professionals Spanish *n* = 90	
	*n* (%)/ M (SD)	*n* (%)/ M (SD)	*n* (%)/ M (SD)	*n* (%)/ M (SD)	$F/\chi^{2}$ (*p*) Post-Hoc/CTR
**PERSONAL AND FAMILY EXPOSURE**	0.65 (0.94)	0.62 (0.90)	0.45 (0.93)	0.81 (0.96)	2.507 (0.084)
**WORK EXPOSURE**	5.08 (3.94)	0 (0)	5.10 (1.87)	8.30 (2.57)	303,490 (<0.001) <0.001 (St < Mx) <0.001 (St < Sp) <0.001 (Mx < Sp)
**TOTAL EXPOSURE**	5.55 (4.09)	0.62 (0.90)	5.55 (2.35)	9.03 (2.66)	240,101 (<0.001) <0.001 (St < Mx) <0.001 (St < Sp) <0.001 (Mx < Sp)
**MCSP**	7.99 (1.02)	7.96 (1.00)	8 (1.03)	7.9 (1.03)	0.082 (0.921)
**PMCS**	10.77 (1.33)	10.48 (1.53)	11.1 (0.93)	10.6 (1.36)	4.241 (0.016) 0.036 (Sp < Mx) 0.014 (St < Mx)
**PIL**	113.24 (15.4)	109.30 (14.1)	123.3 (12.6)	109.0 (14.9)	21.397 (<0.001) <0.001 (Sp < Mx) <0.001 (St < Mx)
**PURPOSE IN LIFE Yes**	113 (55.4)	27 (48.2)	**47 (79.7)**	**39 (43.8)**	20.054 (<0.001) (−1.3/4.4/−2.9)
**GACS-24**	136.7 (17.4)	133.2 (15.6)	145.7 (16.9)	132.9 (16.7)	12.198 (<0.001) <0.001 (Sp < Mx) <0.001 (St < Mx)
**BAI**	8.15 (8.14)	8.85 (6.99)	6.71 (7.95)	8.67 (8.88)	1319 (0.270)
**ANXIETY Yes**	86 (42.6)	**30 (53.6)**	**18 (30.5)**	38 (43.7)	6.327 (0.042) (2.0/−2.2/0.3)
**BDI-II**	7.50 (7.09)	8.80 (5.82)	4.45 (5.63)	8.69 (8.07)	8.177 (<0.001) <0.001 (Mx < St) 0.001 (Mx < Sp)
**DEPRESSION Yes**	35 (17.2)	9 (16.1)	7 (11.9)	19 (21.3)	2.309 (0.315)
**ACUTE STRESS**	7.06 (6.30)	6.07 (3.61)	5.71 (5.49)	8.59 (7.70)	4.852 (0.009) 0.024 (Mx < Sp) 0.024 (St < Sp)
**ACUTE STRESS Yes**	45 (22.1)	8 (14.3)	8 (13.6)	**29 (32.6)**	10.182 (0.006) (−1.6/−1.9/3.2)
**DAST-10**	0.11 (0.45)	0.17 (0.54)	.05 (.22)	0.11 (0.50)	1.123 (0.327)
**DRUGS Yes**	15 (7.4)	7 (12.5)	3 (5.1)	5 (5.6)	3.017 (0.221)
**AUDIT**	2.69 (2.58)	3.51 (3.28)	2.37 (2.53)	2.37 (1.94)	4.065 (0.019) Not significant
**ALCOHOL Yes**	18 (8.9)	7 (12.5)	4 (6.8)	7 (8.0)	1.299 (0.522)
**PSYCHOPATHOLOGY**	26.01 (19.82)	28.42 (14.65)	20.30 (17.89)	28.43 (23.25)	3.569 (0.030) 0.024 (Mx < St)
**MENTAL DISORDER Yes**	114 (55.6)	**39 (69.6)**	**23 (39.0)**	52 (57.8)	11.256 (0.004) (2.5/−3.0/0.6)

CTR: corrected typified residuals; those less than −1.96 or greater than 1.96 were considered significant. The groups from among the categorical variables in which the CTRs were significant are shown in bold. When the post-hoc tests in the quantitative variables were significant we have indicated between which groups the differences occurred. St: Students; Mx: Mexican professionals; Sp: Spanish professionals.

**Table 3 ijerph-18-07235-t003:** Odds ratios from unadjusted and adjusted generalized linear models predicting dependent variables.

RESPONSE	PREDICTORS	UOR 95% CI	*p*-Value	AOR 95% CI	*p*-Value
**BAI**	Personal and family exposure	**3.071** **1.934, 4.207**	<0.001	**1.959** **0.719, 3.200**	0.002
	Work exposure	0.072 −0.220, 0.364	0.627	0.042 −0.233, 0.317	0.766
	Total exposure	0.249 −0.033, 0.532	0.084	-	-
	MCSP	0.589 −0.608, 1.785	0.333	1.060 −0.173, 2.293	0.092
	PMCS	−0.047 −0.924, 0.831	0.917	0.508 −0.368, 1.384	0.256
	PIL	**−0.216** **−0.283, −0.149**	<0.001	**−0.166** **−0.274, −0.057**	0.003
	GACS-24	**−0.148** **−0.211, −0.085**	<0.001	−0.055 −0.148, 0.039	0.252
**BDI-II**	Personal and family exposure	**2.434** **1.417, 3.451**	<0.001	**1.308** **0.351, 2.265**	0.007
	Work exposure	0.082 −0.169, 0.333	0.521	0.040 −0.170, 0.250	0.708
	Total exposure	0.198 −0.050, 0.447	0.116	-	-
	MCSP	−0.106 −1.106, 0.894	0.834	0.941 −0.015, 1.896	0.054
	PMCS	−0.406 −1.159, 0.347	0.289	0.125 −0.554, 0.804	0.718
	PIL	**−0.272** **−0.324, −0.221**	<0.001	**−0.206** **−0.289, −0.123**	<0.001
	GACS-24	**−0.207** **−0.256, −0.158**	<0.001	**−0.084** **−0.156, −0.012**	0.022
**ACUTE STRESS**	Personal and family exposure	**2.218** **1.334, 3.103**	<0.001	**1.181** **0.247, 2.116**	0.013
	Work exposure	**0.343** **0.125, 0.561**	0.002	**0.320** **0.115, 0.525**	0.002
	Total exposure	**0.424** **0.215, 0.634**	<0.001	-	-
	MCSP	0.407 −0.491, 1.305	0.372	0.662 −0.271, 1.595	0.164
	PMCS	0.003 −0.659, 0.665	0.992	0.326 −0.336, 0.989	0.334
	PIL	**−0.149** **−0.201, −0.097**	<0.001	**−0.147** **−0.228, −0.066**	<0.001
	GACS-24	**−0.101** **−0.149, −0.053**	<0.001	−0.003 −0.073, 0.067	0.929
**DAST-10**	Personal and family exposure	−0.012 −0.081, 0.058	0.743	−0.004 −0.083, 0.074	0.912
	Work exposure	−0.008 −0.025, 0.008	0.303	−0.006 −0.023, 0.011	0.502
	Total exposure	−0.008 −0.024, 0.009	0.356	-	-
	MCSP	−0.005 −0.069, 0.059	0.880	0.002 −0.077, 0.080	0.968
	PMCS	0.006 −0.043, 0.056	0.796	0.002 −0.054, 0.058	0.952
	PIL	**−0.006** **−0.010, −0.002**	0.003	**−0.009** **−0.016, −0.002**	0.013
	GACS-24	−0.001 −0.004, 0.003	0.704	0.003 −0.003, 0.009	0.261
**AUDIT**	Personal and family exposure	0.270 −0.121, 0.661	0.174	0.332 −0.105, 0.768	0.137
	Work exposure	**−0.132** **−0.222, −0.043**	0.004	**−0.161** **−0.257, −0.065**	0.001
	Total exposure	**−0.119** **−0.208, −0.029**	0.009	-	-
	MCSP	−0.276 −0.656, 0.105	0.155	−0.103 −0.539, 0.333	0.643
	PMCS	−0.160 −0.442, 0.123	0.267	−0.012 −0.322, 0.298	0.940
	PIL	−0.018 −0.041, 0.006	0.140	0.010 −0.028, 0.048	0.608
	GACS-24	**−0.024** **−0.045, −0.003**	0.023	**−0.039** **−0.071, −0.006**	0.021
**PSYCHOPATHOLOGY**	Personal and family exposure	**8.144** **5.386, 10.903**	<0.001	**4.995** **2.219, 7.771**	<0.001
	Work exposure	0.267 −0.449, 0.983	0.463	0.168 −0.447, 0.783	0.592
	Total exposure	**0.709** **0.013, 1.405**	0.046	-	-
	MCSP	0.742 −2.123, 3.608	0.610	2.534 −0.225, 5.293	0.072
	PMCS	−0.282 −2,454, 1.891	0.798	0.902 −1.058, 2.862	0.367
	PIL	**−0.655** **−0.813, −0.498**	<0.001	**−0.513** **−0.756, −0.271**	<0.001
	GACS-24	**−0.480** **−0.626, −0.334**	<0.001	−0.180 −0.389, 0.030	0.093

UOR: unadjusted odds ratio. AOR: adjusted odds ratio, adjusted according to the other predictor variables plus age, sex, religiosity, psychological or psychiatric treatment during the pandemic, physical illness, and psychiatric history. The significant odd ratios are shown in bold.

**Table 4 ijerph-18-07235-t004:** Significant odds ratios of the logistic regression models predicting the presence of mental disorders.

RESPONSE	PREDICTORS	OR (95% CI)	*p*-Value
**Anxiety disorder**	Personal and family exposure	1.662 (1.090, 2.533)	0.018
	MCSP	1.650 (1.102, 2.472)	0.015
	GACS-24	−0.965 (−0.940, −0.989)	0.005
	Age	−0.958 (−0.927, −0.990)	0.010
	Male gender	−0.371 (−0.154, −0.895)	0.027
	Psychological/psychiatric treatment during the pandemic	−0.080 (−0.014, −0.468)	0.005
**Depressive disorder**	Personal and family exposure	1.968 (1.235, 3.136)	0.004
	PIL	−0.931 (−0.900, −0.962)	<0.001
**Acute stress disorder**	Personal and family exposure	1.911 (1.227, 2.977)	0.004
	MCSP	1.753 (1.097, 2.801)	0.019
	PIL	−0.949 (−0.919, −0.980)	0.001
	Religiosity	−0.229 (−0.082, −0.642)	0.005
	Psychological/psychiatric treatment during the pandemic	−0.080 (−0.015, −0.421)	0.003
	Psychiatric history	4.190 (1.133, 15.498)	0.032
**Problematic drug use**	Psychiatric history	−0.289 (−0.090, −0.928)	0.037
**Problematic alcohol use**	-	-	-
**Any mental disorder**	Personal and family exposure	1.858 (1.203, 2.870)	0.005
	MCSP	1.682 (1.151, 2.458)	0.007
	GACS-24	−0.968 (−0.946, −0.991)	0.007
	Age	−0.970 (−0.941, −0.999)	0.044

Note: predictor variables plus age, sex, religiosity, psychological/psychiatric treatment during the pandemic, physical illness, and psychiatric history were all entered into all the models.

## Data Availability

The data that support the findings of this study are available on request from the corresponding author. The data are not publicly available due to privacy or ethical restrictions.

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
