# Peer review of "“Healthcare Kamikazes” during the COVID-19 Pandemic: Purpose in Life and Moral Courage as Mediators of Psychopathology"

_ijerph, 2021, doi:10.3390/ijerph18147235_

Round 1
Reviewer 1 Report
Re: ijerph-1262952, “Healthcare kamikazes” during the COVID-19 pandemic: purpose in life and moral courage as mediators of psychopathology
The study probed the nature of mental health among medical professions in two nationalities: Spain and Mexico. The topic is important, and the instruments is sound. Also, the information in this manuscript may provide important information for readers in Latin culture. Nevertheless, I have some comments listed below:
- As this is a convenience sample in limited sites, a brief introduction of the study site is needed for readers.
- The path model should be trimmed extensively. I suggest a clear hypothesis testing the final model would be better than data-driven approach with theoretical purpose.
- The test for buffer effect is not completed yet. A simple main effect for the buffer effect and a figure is needed. Please refer to classical reading of Cohen. Cohen, S., & Wills, T. A. (1985). Stress, social support, and the buffering hypothesis. Psychological Bulletin, 98(2), 310–357. https://doi.org/10.1037/0033-2909.98.2.310
Author Response
Reviewer 1
Point 1: As this is a convenience sample in limited sites, a brief introduction of the study site is needed for readers.
Response 1: We have included a brief description about the hospitals involved in the study: Consorci Hospitalari Provincial de Castelló which is the second biggest hospital in the city, responsible, among others, of the mental health, oncology and ophthalmology departments in the province of Castelló (…) Hospital Psiquiátrico de Campeche, the main mental health institution in the province of Campeche. Thank you for the appreciation.
Point 2: The path model should be trimmed extensively. I suggest a clear hypothesis testing the final model would be better than data-driven approach with theoretical purpose.
Response 2: We have reduced the number of models to 4 to focus on testing two hypotheses, adding in methodology: Finally, the data were modeled using PROCESS v3.4 for SPSS [22] to test two hypotheses: 1) personal and family exposure increases anxiety, depression, acute stress and total psychopathology; and 2) the purpose in life has a buffering effect, modulates the relationship between personal and family exposure and psychopathology. We have also adapted results and the discussion to these changes. Thank you very much for the suggestion, we believe that this way the models are easier to understand and give more specific information.
Point 3: The test for buffer effect is not completed yet. A simple main effect for the buffer effect and a figure is needed. Please refer to classical reading of Cohen. Cohen, S., & Wills, T. A. (1985). Stress, social support, and the buffering hypothesis. Psychological Bulletin, 98(2), 310–357. https://doi.org/10.1037/0033-2909.98.2.310
Response 3: We have focused the path models on the buffering effect of purpose in life, thus reflecting it in Figure 1, the results and the discussion. We have referenced this classic article in the discussion: Considering all the above, there is evidence that high levels of PIL played a protective role (the buffering model described by Cohen and Wills) [30] in the effect that personal and family exposure to the virus had on overall pathology, anxiety and acute stress.
Reviewer 2 Report
The paper deals with a relevant topic, i.e. the psychological well-being of healthcare workers confrontated with the SARS-CoV-2 pandemic. The study is timely and the manuscript is interesting and well written. I was truely happy to be choosen to serve as reviewer.I do not have much comments to share, but I suggest: 1) to change "X ̅" to a more standard abbreviation (I suppose "M" for mean could be ok) 2) to avoid " in Conflicts of interest statement.
Author Response
Reviewer 2
Point 1: to change "X ̅" to a more standard abbreviation (I suppose "M" for mean could be ok)
Point 2: to avoid " in Conflicts of interest statement.
Response 1&2: Thank you for your kind advices. We have add both of them in the manuscript.
Reviewer 3 Report
I would like to see a much more in-depth discussion of the terms "purpose in life" and "moral courage" and how they might relate to psychology generally and I would like to see a more robust discussion of the findings. Why would purpose in life differ from moral courage in the outcomes. This is discussed by not convincingly. The paper would benefit from some case reports with individual psychology to illustrate more clearly what the finding might mean.
Author Response
Reviewer 3
Point 1: I would like to see a much more in-depth discussion of the terms "purpose in life" and "moral courage" and how they might relate to psychology generally and I would like to see a more robust discussion of the findings. Why would purpose in life differ from moral courage in the outcomes. This is discussed by not convincingly.
Response 1: We have added a more in-depth explanation of purpose in life from the point of view of positive psychology and its link with the psychopathology: These results may be explained by the fact that PIL is included in the logotherapy which holds that life can have purpose and sense even in the most impoverished circumstances, making people more resilient in terms of surviving harsh conditions. Both terms are related to the positive psychology and the salutogenesis framework, which states that people who view their life as having positive influence on their health, may stay well and even be able to improve adaptive coping in stressful situations. This framework assumes that people have resources available (biological, material, and psychosocial) that enable them to construct coherent life experiences. Failure to do so could explain their psychopathology [30].
Unfortunately, moral courage is still an understudied dimension and there is not much information about it apart from moral distress. However, we have tried to extend a little more why the moral courage would lead to an increased psychopathology: causing the professional to be prevented from acting in the way he or she considers correct and increasing their psychopatology [32].
We have emphasized the differences between purpose in life and moral courage, which seems to be responsible for the different outcomes: Despite their similarities, intrinsic factors that determine PIL and moral courage are not the same: while PIL depends on life perception and goals, moral courage is influenced by ethical principles and moral obligation, which would explain the differences between both and the derived psychopathology.
Point 2: The paper would benefit from some case reports with individual psychology to illustrate more clearly what the finding might mean.
Response 2: It is true that an idiographic approach would provide more detailed information on how purpose in life and moral courage influence each professional, but unfortunately we have not collected individualized and qualitative information by having designed the study from a nomothetic perspective.
Thank you for your interesting and valuable comments and approach.
Round 2
Reviewer 3 Report
None